# Targeting Isocitrate Dehydrogenase (IDH) in Solid Tumors: Current Evidence and Future Perspectives

**DOI:** 10.3390/cancers16152752

**Published:** 2024-08-02

**Authors:** Francesca Carosi, Elisabetta Broseghini, Laura Fabbri, Giacomo Corradi, Riccardo Gili, Valentina Forte, Roberta Roncarati, Daria Maria Filippini, Manuela Ferracin

**Affiliations:** 1Medical Oncology Unit, IRCCS Azienda Ospedaliero-Universitaria di Bologna, 40138 Bologna, Italy; francesca.carosi3@studio.unibo.it (F.C.); laura.fabbri21@studio.unibo.it (L.F.); giacomo.corradi@studio.unibo.it (G.C.); 2IRCCS Azienda Ospedaliero-Universitaria di Bologna, 40138 Bologna, Italy; elisabetta.broseghini@aosp.bo.it; 3Medical Oncology Unit, IRCCS Ospedale Policlinico San Martino, 16132 Genova, Italy; giliriccardo.rg@gmail.com; 4Diagnostic Imaging Unit, Department of Biomedicine and Prevention, University of Rome Tor Vergata, 00133 Rome, Italy; valentina.forte.01@students.uniroma2.eu; 5Istituto di Genetica Molecolare “Luigi Luca Cavalli-Sforza”, Consiglio Nazionale delle Ricerche (CNR), 40136 Bologna, Italy; roberta.roncarati@cnr.it; 6Department of Medical and Surgical Sciences (DIMEC), University of Bologna, 40126 Bologna, Italy

**Keywords:** IDH1, IDH2, biomarker, glioma, cholangiocarcinoma, chondrosarcoma, targeted treatment, clinical trial

## Abstract

**Simple Summary:**

Isocitrate dehydrogenase 1 and 2 (IDH1 and IDH2) are essential metabolic enzymes involved in the tricarboxylic acid (TCA) cycle. Several mutations in *IDH* genes have recently been described in many solid tumors, including glioma, cholangiocarcinoma, and chondrosarcoma. These mutations lead to neomorphic enzymatic activity affecting cancer pathogenesis. This review aims to summarize the diagnostic and prognostic role of *IDH* mutations and to provide an overview of the actual IDH inhibitor-based therapies used in various solid malignancies, outlining the findings of the most recent clinical trials and searching for future perspectives.

**Abstract:**

The isocitrate dehydrogenase 1 and 2 (IDH1 and IDH2) enzymes are involved in key metabolic processes in human cells, regulating differentiation, proliferation, and oxidative damage response. IDH mutations have been associated with tumor development and progression in various solid tumors such as glioma, cholangiocarcinoma, chondrosarcoma, and other tumor types and have become crucial markers in molecular classification and prognostic assessment. The intratumoral and serum levels of D-2-hydroxyglutarate (D-2-HG) could serve as diagnostic biomarkers for identifying IDH mutant (IDHmut) tumors. As a result, an increasing number of clinical trials are evaluating targeted treatments for IDH1/IDH2 mutations. Recent studies have shown that the focus of these new therapeutic strategies is not only the neomorphic activity of the IDHmut enzymes but also the epigenetic shift induced by IDH mutations and the potential role of combination treatments. Here, we provide an overview of the current knowledge about IDH mutations in solid tumors, with a particular focus on available IDH-targeted treatments and emerging results from clinical trials aiming to explore IDHmut tumor-specific features and to identify the clinical benefit of IDH-targeted therapies and their combination strategies. An insight into future perspectives and the emerging roles of circulating biomarkers and radiomic features is also included.

## 1. Introduction

### 1.1. Physiological Role of IDH Enzymes in Cell Metabolism

The IDH family is composed of three different enzymes, IDH1, IDH2, and IDH3, that are all involved in cell metabolism and catalyze the same reaction, namely the oxidative decarboxylation of isocitrate (ICT) into α-ketoglutarate (α-KG). Despite this functional overlap, their role in cellular metabolism is non-redundant. The cytoplasm and peroxisomes are the most frequent localizations of the IDH1 enzyme, while the IDH2 and IDH3 enzymes are located in the mitochondrial matrix, where they are responsible for one of the steps in the tricarboxylic acid (TCA) cycle (Figure 1).

IDH1 and IDH2 activation is mediated by homodimer formation. They share the same molecular mechanism: the conversion of ICT into α-KG, which involves the reduction of the cofactor nicotinamide adenine dinucleotide phosphate (NADP) [1,2,3]. Differently from IDH1 and IDH2, IDH3 has a heterotetrameric structure in its active form, derived from the union of two catalytic subunits (encoded by the *IDH3A* gene) and two regulatory ones, IDH3B and IDH3C. Specifically, through the reduction of NAD, IDH3 converts isocitrate into α-KG [4,5,6]. In contexts such as cell hypoxia, the IDH enzymes are also able to catalyze the opposite reaction and obtain ICT from α-KG due to glutamate deamination. ICT can supply the Krebs cycle and fatty acid production [7,8]. The NADH and NADPH generated in these reactions are used in the electron transport chain to neutralize oxygen-reactive species or as substrates in cholesterol and fatty acid synthesis [6,9].

### 1.2. Tumorigenesis Induced by IDH Mutations

The *IDH1* and *IDH2* genes are characterized by a high rate of gain-of-function mutations across various tumor types. These genes often present with missense mutations particularly concentrated in hotspot regions. The most frequent *IDH1* mutations occur on the residue R132, including R132H, R132C, R132S, R132G, and R132L. For *IDH2*, R140 and R172 are the most frequent mutation points, which include R140Q, R172G, R172K, R172M, R172S, and R172T substitutions [10,11,12,13,14,15]. These amino acid residues are strategically positioned within the binding site of the natural enzymatic substrate, so their mutation alters the three-dimensional conformation and binding capability, resulting in a neomorphic gain of function [16]. The mutations lead to a decreased binding affinity to ICT, while the affinity to NADPH increases. The different conformation and the altered binding affinity result in a loss of catalysis of ICT oxidation and a gain in the catalysis of a partial reverse reaction, in which α-KG is reduced to (R)-2-hydroxyglutarate [(R)-2-HG] and not further carboxylated. Furthermore, IDH mutations are associated with an altered ratio of the two enantiomers of 2-HG (D-2-HG and L-2-HG), inducing an increase in D-2-HG levels [17,18,19] (Figure 1).

The altered physiological and correct catalytic reaction of IDH1 and IDH2, which is caused by a mutation in these genes, leads to the accumulation of 2-HG, which becomes one of the most concentrated tumoral cell elements [20]. 2-HG has a strong structural similarity to α-KG, and when it is more abundant in the cellular environment, it can replace α-KG in binding to its classical substrates [21,22]. This is reflected on the inhibition of α-KG-dependent dioxygenases [21,22,23], such as the TET-DNA demethylases, and the Jumonji family histone demethylases (KDMs) with consequent DNA and histone hypermethylation and a block in cell differentiation, where high levels of D-2-HG in the interstitial fluid of tumor cells lead to impairs T cell proliferation and their cytotoxicity mechanisms [21,23,24,25,26,27,28,29,30,31,32]. Other demethylases that act in the DNA repair machinery (e.g., ALKBH2/3), the HIF1α signaling pathway (e.g., asparaginyl hydroxylase and the prolyl hydroxylase domain proteins), and fatty acid production (e.g., RNA N6-methyladenosine (m6A) demethylase) are targets of 2-HG inhibition [18,33,34]. 2-HG affects the activity of transaminases, including the branched-chain aminotransferases BCAT1 and BCAT2, which are fundamental for the degradation of branched amino acids [35]. 

*IDH* mutations have been firstly identified in colorectal cancer and glioblastoma [36,37] and lately associated with the occurrence of acute myeloid leukemia (AML) (~20% of cases) [38,39,40,41]; angioimmunoblastic T lymphoma (32%); and various solid tumors including low-grade glioma and secondary glioblastoma (80%) [10,42,43], cholangiocarcinoma (20%) [44,45], chondrosarcoma (50–80%) [46], and sinonasal undifferentiated carcinoma (SNUC) (49–82%) [47,48] (Figure 2 and Table 1).

This review aims to illustrate the diagnostic and prognostic role of *IDH* mutations. We provide an overview of the actual IDH inhibitors evaluated across various solid malignancies, outlining the results of the most recent clinical trials and looking to future perspectives. 

## 2. *IDH* Mutation and IDH Targeted Therapies in Various Solid Malignancies

### 2.1. Glioma

The value of *IDH1* and *IDH2* gene mutations in glioma was investigated in 2009 by Yan et al. [10]. The authors discovered that an *IDH* mutation was the only genetic alteration found in grade 2 or 3 astrocytomas and oligodendroglioma and could be involved in the early development of glioma. The main mutations identified were the substitution of an Arginine with a Histidine at residue 132 of IDH1 (R132H) and the same substitution at residue 172 of IDH2 (R172H) [10].

Mutated *IDH1* has been described as responsible for the remodeling of the methylome in glioma. Specifically, IDH1 established the CpG island methylator phenotype (CIMP), characterized by extensive epigenetic aberrations and a powerful determinant of tumor pathogenicity [29].

Over the years, it has been recognized that mutated forms of *IDH1* and *IDH2* may contribute to tumor development and serve as diagnostic markers. Consequently, the presence of *IDH* mutations was incorporated into the WHO glioma classification in 2016 [49]. Regardless of grade and treatment, the presence of *IDH* mutations is associated with a better prognosis. Indeed, low-grade, *IDH* wild-type gliomas are potentially as aggressive as glioblastomas with a similar prognosis [49,50]. Apparently, there is no evidence of survival outcome differences between IDH1mut and IDH2mut glioma [51]. Given the potential prognostic role of IDH, in 2021, the WHO classification was further revised to categorize *IDH*-mutated tumors as low-grade gliomas (LGGs) [52].

In addition to the diagnostic and prognostic role of IDH mutations in glioma, IDH was evaluated as a therapeutic target. 

An inhibitor, namely AGI-5198, was specifically developed against the mutation R132H IDH1, and its efficacy was evaluated in 2013. This inhibitor blocks the enzyme’s ability to produce 2-HG, due to its structural similarity to α-KG and competing with α-KG-dependent dioxygenases. This inhibitor leads to the demethylation of histone H3K9me3 and consequently induces the expression of genes involved in glioma differentiation. A pharmacological blockade of mutated IDH1 impaired the growth of IDH1mut but not IDH1-wild-type, glioma cells [24].

In 2014, Ivosidenib (AG-120) was evaluated in IDHmut tumors, including glioma, showing better survival for the non-enhancing tumors, namely LGG, compared to the enhancing gliomas [53,54]. In 2015, Vorasidenib (AG- 881) [55], which is a pan-IDH inhibitor (IDH1/IDH2 inhibitor), was tested, showing a favorable safety profile, an objective response rate (ORR) of 18.2%, and a median progression-free survival (PFS) of 36.8 months [56]. In 2017, Ivosidenib and Vorasidenib were evaluated in a phase I randomized trial [57] in perioperative patients with recurrent low-grade gliomas. The primary endpoint was the reduction in 2-HG concentration in glioma tissues. The randomized treatments were Vorasidenib, Ivosidenib, or no treatment before surgery. The 2-HG concentration was reduced by 92.6% in patients treated with Vorasidenib and by 91.1% in patients treated with Ivosidenib [58]. Vorasidenib, which showed a more consistent 2-HG suppression and brain penetrance, advanced to a phase III trial, namely the INDIGO trial [59]. Patients with residual or recurrent grade 2 glioma characterized by an IDH1 or IDH2 mutation were randomly assigned to receive Vorasidenib or a placebo. The last update showed that PFS was significantly improved in patients who received Vorasidenib (27.7 months for the Vorasidenib group vs. 11.1 months with the placebo group) [60]. The synergic effect of chemotherapy (azacytidine) and IDH inhibitors (ivosidenib) was evaluated in IDH1mut AML patients, and this combination is now approved by the FDA for patients not eligible for intensive induction chemotherapy; a similar therapeutic strategy could be further evaluated in IDHmut gliomas [61].

Other IDH inhibitors are currently under evaluation. Enasidenib is an IDH2mut inhibitor evaluated in a basket trial in 2014 [62]; however, the results on the glioma cohort have not yet been published. Another drug currently being evaluated is Olutasidenib (FT-2102). In 2018 a basket trial [63], that included gliomas, Olutasidenib 150 mg was tested twice a day as a single agent or in combination with Azacitidine, a pyrimidine analogue, in patients with relapsed/refractory IDH1mut R132X gliomas. A disease control rate (DCR) in 48% of cases with acceptable tolerability was revealed [64,65]. Another IDH1mut inhibitor is DS-1001b, a selective IDH1 inhibitor, mostly active on IDH1 R132H and IDH1 R132C mutations. A phase II study assessed the efficacy and safety of DS-1001b in patients with chemotherapy- and radiotherapy-naive IDH1-mutated WHO grade 2 gliomas [66]. Finally, two phase I studies analyzed two different IDH1 inhibitors, namely IDH305 and BAY1436032. One of them studied the use of IDH305 in patients with advanced malignancies that present IDH1R132 mutations [67], while the other trial showed that the use of BAY1436032 in advanced solid tumors significantly decreases the level of 2-HG in serum and prolongs the survival of human astrocytomas [68,69].

Table 2 summarizes the features of completed and ongoing clinical trials that evaluate IDH inhibitors for the treatment of glioma, and Figure 3 shows the chemical structures of the drug tested.

**Table 2 cancers-16-02752-t002:** List of completed and ongoing clinical trials that evaluate(d) IDH inhibitors for the treatment of glioma. The IDH mutation types and their frequency are also reported.

Glioma					
IDH Mutation Types				IDH1 R132H, IDH2 R172H	
Frequency of IDH Mutations				>80% (Grade 2 and Grade 3 Glioma) 73% (Secondary Glioblastoma) 3.7% (Primary Glioblastoma)	
Trial Name	Phase	Year	Drug Tested	Target Population	Outcome Measure
**Clinical trials—Completed**					
NCT02073994 [54]	I	2014–2024	Ivosidenib	IDH1mut advanced solid tumors	ORR 2.9%, mPFS 13.6 mo
NCT02481154 [56]	I	2015–2024	Vorasidenib	IDH1mut or IDH2mut advanced solid tumors	ORR 18%, mPFS 36.8 mo
NCT03343197 [58]	I	2019	Vorasidenib, ivosidenib	Recurrent low-grade glioma	Reduced concentration of 2-HG (~92%)
NCT03684811 [70]	Ib/II	2018–2022	Olutasidenib	Relapsed/refractory IDH1mut advanced solid tumors	DCR 48%
NCT02746081 [68]	I	2016	BAY1436032	IDH1mut advanced solid tumors	ORR 11%
NCT03030066 [71]	I	2017	DS-1001b	Recurrent/progressive IDH1mut glioma	mPFS 10.4 mo
NCT02273739	I/II	2014–2021	Enasidenib	IDH2mut advanced solid tumors	NA
NCT04164901 (INDIGO trial) [59]	III	2019–2023	Vorasidenib vs. placebo	Recurrent/residual grade 2 glioma with IDH1 or IDH2 mutations	mPFS 27.7 vs. 11.1 mo, TTNT NA vs. 17.8 mo
**Clinical trials—Ongoing**					
NCT04458272 [66]	II	2020	DS-1001b	IDH1mut grade 2 glioma (CHT and RT naive)	
NCT04762602 [72]	I	2021	HMPL-306	IDHmut solid tumors	
NCT04521686	I	2020	LY3410738	IDH1mut or IDH2mut advanced solid tumors	
NCT02381886	I	2015	IDH305	IDH1R132-mut advanced solid tumors	
NCT06161974	II	2024	Olutasidenib	IDH1mut high-grade glioma	

2-HG: 2-hydroxyglutarate; ORR: objective response rate; DCR: disease control rate; mPFS: median progression-free survival; TTNT: time to next treatment; NA: not available; mo: months. CHT: chemotherapy; RT: radiotherapy.

In conclusion, considering that IDH mutations are a foundational event in glioma progression, targeting IDH mutations at an early stage of the disease is crucial to halt disease progression and prevent the acquisition of additional genetic alterations that could reduce the effectiveness of IDH inhibitors [73].

### 2.2. Cholangiocarcinoma

Biliary tract cancers (BTCs) are a rare oncological entity accounting for less than 1% of all tumors. Among these cancers, cholangiocarcinoma (CCA) is the second most frequent hepatic neoplasia. CCAs are classified, based on their origin, into intrahepatic cholangiocarcinoma (iCCA), perihilar cholangiocarcinoma (pCCA), and distal cholangiocarcinoma (dCCA) [74]. 

Surgery, followed by adjuvant chemotherapy or not, is the preferred treatment for localized disease, but recurrence rates are elevated and CCAs tend to have a poor prognosis [75,76,77]. In the case of advanced or metastatic disease, recently, molecular analysis started to become an essential part of the diagnostic work-up, leading to a progressive change in the therapeutic strategies [74,78].

Targetable mutations are detectable in over 40% of iCCA, with the most frequent being *IDH1/IDH2* mutations (20–30%, mostly involving IDH1), FGFR-2 fusions (10–15%), HER-2 amplifications (5–15%), BRAF V600E mutations (4–5%), and neurotrophic tyrosine receptor kinase (NTRK) fusions (about 1%) [79,80,81,82]. The most frequent mutation occurs in residual 132 (R132) for IDH1 and residual 172 (R172) for IDH2. A correlation between IDH1/IDH2 mutated phenotype and clinical parameters has yet to be established, and the available data are controversial. The prognostic role of IDH1 or IDH2 mutations in CCA remains unclear: some studies suggested IDH1 or IDH2 mutations as positive prognostic factors, showing an improvement in overall survival (OS) and disease-free survival (DFS) in CCA patients after surgery [83,84,85], whereas other research has not found a significant association between IDH mutations and OS or DFS [86,87,88]. 

The IDH1 inhibitor Ivosidenib has been shown to increase survival outcomes compared to a placebo in pre-treated IDH1mut iCCA patients in the phase III ClarIDHy trial. Specifically, PFS in patients who received Ivosidenib was 2.7 months compared to 1.4 months for the placebo groups, and the median OS was 10.8 months vs. 9.7 months in the Ivosidenib and placebo groups, respectively. These results, in addition to a good safety profile, have led to the FDA approval of Ivosidenib with this specific indication [89,90]. However, this phase III trial has some important limitations. First of all, the possibility of crossover for patients in radiological progression in the placebo arm has introduced a statistical bias. Secondly, the study design is built upon the randomization of patients to receive a placebo or Ivosidenib, so a comparison between targeted therapy and II-line FOLFOX chemotherapy is still lacking. Ivosidenib is commercialized as oral capsules, and the recommended dosage is 500 mg once daily [91]. The most commonly reported adverse events are nausea, diarrhea, and fatigue, with a low rate of treatment discontinuation.

To discover therapeutic agents capable of providing a prolonged disease response, scientific efforts led to the identification of the IDH1 inhibitor LY3410738, which has shown efficacy against the D279N mutation [92]. This inhibitor is currently being evaluated in clinical trials (NCT04603001 [93], NCT04521686 [94]). To overcome the resistance induced by an isoform switching mutation, the possibility to use the anti-IDH2 Enasidenib (yet approved for AML-resistant clones), the association between Enasidenib and Ivosidenib (anti-IDH1 plus anti-IDH2), and the use of dual inhibitors like Vorasidenib have been explored as promising strategies in AML and glioma, while in iCCA, the data are still insufficient [56,95].

Recently, a phase Ib/II (NCT03684811) study evaluated Olutasidenib (FT-2102) as a monotherapy or in combination with other antitumoral drugs in various tumors. For BTCs, the study design [63] included two different cohorts, one for hepatobiliary tumors with the administration of FT-2102 plus nivolumab and the other for iCCA with FT-2102 plus gemcitabine/cisplatin. Limited activity with ORRs of 12.5% and 0%, respectively, was demonstrated.

Although not yet recruiting, the NCT05814536 [96] trial will assess the safety and clinical efficacy of Safusidenib (AB-218), a selective IDH1 inhibitor, with the advantage of oral administration, in patients with IDH1mut advanced CCA and other solid tumors.

In addition to IDH inhibitors, another research line has focused on the potential role of the multi-kinase inhibitor Dasatinib, exploiting its capability to block SRC tyrosine kinase intracellular signaling. SRC kinases have a pro-oncogenic role favoring proliferation, distant migration, and infiltration of tumoral cells and promoting angiogenesis [97]. Saha et al. demonstrated that IDHmut cellular lines and xenograft models were hyper-sensitive to Dasatinib, and they hypothesized that the high response to the drug was linked to a dependence of tumoral cells on the SRC kinase pathway for self-maintenance. Despite these encouraging premises, a recent phase II trial evaluating Dasatinib in patients with advanced IDH1 or IDH2mut iCCA who underwent at least one prior chemotherapy platinum-based regimen [98] showed an ORR of 0%, a PFS of 8.4 weeks, and an OS of 37.9 weeks, with a negative toxicity profile. These results suggest that the activity of Dasatinib could be enhanced by the association with another drug, such as Ivosidenib, which synergizes with Dasatinib. The combination strategy of Dasatinib and Ivosidenib is likely to be evaluated shortly in the evolving scenario of targeted therapies in iCCA [99].

IDH1/IDH2mut cells require the presence of α-KG to synthesize 2-HG and to sustain their metabolism. α-KG can be a product of two different metabolic routes, glycolysis or glutaminolysis. The reduction of IDH function in mutated cells determines a down-regulation of the glycolytic process; in this case, α-KG concentration depends almost completely on the glutamate dehydrogenases that convert glutamate to α-KG [100]. 

This reaction can be inhibited by the common drug chloroquine used in malaria treatment and by the oral hypoglycemic drug metformin [101,102,103,104,105]. Drugs like metformin and chloroquine can increase metabolic stress because they interfere with the Krebs cycle and, in this way, can deplete the tumoral microenvironment of nutrients. The association of metformin and chloroquine is still under evaluation [106] in patients with IDH1/IDH2mut iCCA, glioma, or chondrosarcoma identified by NGS or 2-HG dosage in circulation, in the tumor, or the DNA sequencing of (circulating) tumor material.

High concentrations of 2-HG in IDHmut neoplasia are strongly correlated with altered DNA repair, homologous recombination (HR) defect, and dissemination of single-strand breaks (SSBs). In detail, the high levels of 2-HG determine the hypermethylation of histone 3 lysine 9 in DNA break sites, and this pattern represents a confounding element for the identification of trimethylation sites that acts as a recruitment signal for homologous repair machinery [107,108,109], namely poly(adenosine diphosphate ribose) polymerase (PARP). This suggests the sensibility of IDHmut cells to PARP inhibitors, such as Olaparib. This vulnerability is the main topic of recruiting trials evaluating the therapeutic potential of PARP-inhibitors, alone or in combination with other drugs in IDHmut CCA [110], NCT03991832 [111], NCT03878095 [112], and NCT03212274 [113]. 

As previously discussed, IDH1 and IDH2 mutations establish an immunologically cold background with low lymphocyte infiltrates at the tumor site; the treatment with Ivosidenib has been proved to be able to recruit CD8+ T cells and restore immune system tumor vulnerability in cholangiocarcinoma [114]. In addition to this, Ivosinedib resistance has been also linked to the increased expression of immunomodulating receptors on neoplastic cells, like PD-L1 and CTLA-4 [115,116]. Recently, the TOPAZ-1 trial established a new standard of care based on chemotherapy and immunotherapy with Durvalumab for patients with advanced biliary tract cancers [117,118]; mounting evidence is now supporting the study of IDH inhibitors in association with immunotherapy for cholangiocarcinoma. On the contrary, in other IDH-mutant tumors such as gliomas, immunotherapy has not met expectations, despite the demonstrated efficacy in preclinical and clinical studies, probably due to the immunosuppressive microenvironment, resulting in drug resistance [119].

Finally, the possibility of exploiting standard chemotherapy in combination with IDH inhibitors represents an intriguing therapeutic avenue. In particular, an active trial is focusing on the safety of cisplatin and gemcitabine with Ivosidenib (Arm A) or with Pemigatinib (Arm B) in patients with advanced cholangiocarcinoma [120]. New perspectives will emerge after the disclosure of definitive results from phase I studies about LY3410738 and HMPL-306, a dual IDH1/IDH2 inhibitor [72,94]. 

Table 3 summarizes the features of completed and ongoing clinical trials that evaluate(d) IDH inhibitors for the treatment of CCA, and Figure 3 shows the chemical structures of the drug tested.

### 2.3. Chondrosarcoma

Chondrosarcoma is the second most common bone tumor. The main type is conventional chondrosarcoma, which includes the central, peripheral, and periosteal subtypes according to the anatomical location of the tumors and originates from the medullary cavity, involving the bones of the pelvis, femur, humerus, and ribs. An adequate surgical excision represents the only curative treatment; if not surgically manageable, poorly effective therapeutic options are currently available [123], especially due to the intrinsic chemo- and radiotherapy resistance of chondrosarcomas.

IDH mutations have been discovered in 50% of conventional and dedifferentiated chondrosarcomas, prevailing in the chondrosarcomas of bone extremities and the skull base (up to 60% of cases). There is not a dominant IDH1 mutation, as observed in other IDH1mut tumors, but the most frequent is R132C, followed by R132G and R132L; on the contrary, IDH2 mutation only involves codon 172 [46,124,125]. Interestingly, 40% of chondrosarcomas that harbor an IDH1 R132C mutation are characterized by a high production of 2-HG [126]. IDH2 mutations are extremely diffused in dedifferentiated chondrosarcomas, helping the differential diagnosis from osteosarcoma [127].

IDH mutations have been also identified in enchondromas, considered as benign precursor lesions of chondrosarcomas [128]; this finding suggests that IDH mutations are early genetic events in the process of carcinogenesis in the chondrogenic lineage [129]. 

Moreover, both IDH-mutated enchondromas and chondrosarcomas are characterized by a typical hypermethylated phenotype involving CpG islands, with the number of methylated genes increasing upon tumor progression. Despite the presence of the altered methylome, in vitro models testing DNA methyltransferase (DNMT) inhibitors (such as decitabine and azacitidine) failed. On the contrary, histone deacetylase (HDAC) inhibitors, especially pan-HDAC inhibitors (Dacinostat, Panobinostat, and more than other 100 compounds) and the class I HDAC inhibitor romidepsin were effective regardless of IDH mutation status and chondrosarcoma subtype [129]. This evidence suggests that the epigenetic mechanism underlying the inhibition of tumor suppressor genes may be independent of the hypermethylated state induced by IDH1/IDH2 mutation. Lately, a combination of DNMT inhibitors and HDAC inhibitors has been tested in vitro, showing promising results, but further studies are needed to demonstrate their efficacy in vivo [130].

Ivosidenib was first evaluated in a phase I study [53] including IDH1mut advanced chondrosarcomas. A significant reduction in tumoral and plasmatic 2-HG levels was observed. Moreover, 65% of patients reached stable disease versus 35% of patients who progressed. The median PFS was 5.6 months, with a good safety profile. Notably, the efficacy of Ivosidenib seemed to be better for patients with conventional chondrosarcomas, who experienced clinical benefit and prolonged disease control (>2.5 years without progression), also if pretreated. This efficacy could be biased by the general indolent behavior of IDH1mut tumors compared to wild-type IDH1 tumors, but retrospective studies had not clearly defined if the IDH1 mutation could have a prognostic role in chondrosarcomas [131,132]. The phase I study was limited by a small number of patients. At the moment, a phase II clinical trial [133] is ongoing to evaluate Ivosidenib in locally advanced, metastatic, or recurrent grade 2 or grade 3 IDH1mut chondrosarcomas; the results of this trial should be available in March 2026.

A novel IDH1 inhibitor molecule, DS-1001b [134], demonstrated the property of inhibiting the growth of IDH1mut chondrosarcoma cells in vitro and in vivo and consequently also 2-HG overproduction due to the IDH1 mutation, similar to Ivosidenib, but with the additional effect of reducing the levels of H3K4me3 and H3K9me3, reversing the epigenetic process induced by 2-HG [135]. Moreover, recent studies showed that IDH1/IDH2mut tumors are affected by defective base excision repair and homologous recombination repair, due to the 2-HG overproduction, which inhibits two dioxygenases, KDM4A and KDM4B, implied in the DNA damage response, and histone hypermethylation. Consequently, the phase II OLAPCO clinical trial [136] was designed to verify the efficacy of PARP inhibition, specifically with Olaparib, in monotherapy or in combination with other target drugs, as a basket trial conducted in several IDHmut tumors, including chondrosarcoma. While other tumors did not have a brilliant response to Olaparib, patients with IDH1mut chondrosarcoma achieved prolonged stable disease or partial response in nearly half of cases [137].

Recently, case reports proved the efficacy of anti-PD1 antibodies, such as Pembrolizumab (KEYNOTE-966) [138], in metastatic conventional chondrosarcomas, leading to a near-complete response and tumor regression [139]. Based on these results, the chondrosarcoma immune tumor microenvironment was investigated and a peculiar “immune exhausted” profile was discovered, typically associated with IDH mutations, high grade, and peritumoral edema [140].

### 2.4. Other Solid Tumors

Recent studies have identified IDH mutations in a subset of SNUCs. IDH1–2 mutations have been detected in nearly 49% of SNUCs and 37.5% of poorly differentiated sinonasal carcinomas, often coexisting with the p53, KIT, or PI3K pathway mutations [141]. 

High-grade carcinomas are more likely to be interested in IDH2 mutations involving the R172 codon, in particular the R172S and R172T variants in 80% of cases [142,143]. As for chondrosarcomas, the IDH mutation induces a hypermethylation profile. This group of patients with IDHmut SNUC presents peculiar histopathological (presence of tumor necrosis and increased mitosis) and clinical (better prognosis and lower propensity for lung metastasis) features [144,145]. According to this evidence, SNUCs have been evaluated in separate categories based on the targetable molecular subtypes with clinical implications [48]. Further studies may be conducted to evaluate IDH2-targeted therapy in this pathological entity.

The IDH1 and IDH2 mutations are also present in 0.9% of colorectal cancer, associated with the BRAF V600E mutation [146], in 0.5% of non-small cell lung carcinoma (NSCLC), co-existing with KRAS mutations [147] and in melanoma, occurring with NRAS mutations [148]. Similarly, IDH2 mutations have been found in solid papillary carcinoma with reverse polarity, a rare breast cancer subtype with unusual histopathological features [127]; in papillary thyroid carcinoma, associated with the development of lymph node metastasis [149]; and in gastric cancer [150]. In these malignancies, IDH mutations interest older patients and high-grade tumors. Interestingly, IDH2 is also significantly decreased in hepatocarcinoma (HCC) tissues, probably promoting the formation of metastasis due to a negative correlation with matrix metallopeptidase 9 (MMP9) [151], with a prognostic and predictive role in HCC patients [152,153]. Also in prostate cancers, tumor progression is led by an integrated signaling between androgen receptors (AR) and the extra-mitochondrial IDH1 activity, suggesting that targeted IDH1 therapies may be a possible therapeutic approach [154,155].

## 3. Future Perspectives

### 3.1. IDH-Related Tissue and Circulating Biomarkers

Since the relevance of *IDH* somatic mutations in solid tumors has increased, the development of a fast and sensitive method to detect IDH mutations is needed, especially to select patients eligible for anti-IDH1 and anti-IDH2 targeted therapies.

Next-generation sequencing (NGS) is a sensitive and specific method to investigate the *IDH* mutational status, but it requires a long running time. Therefore, *IDH* mutations could be identified by immunohistochemistry [156], Sanger DNA sequencing [157], or quantitative PCR. Recently, droplet digital PCR (ddPCR) multiplex assays were tested as an alternative to NGS, detecting 99.8% and 98.9% of *IDH1* and *IDH2* mutations, respectively, according to the COSMIC (Catalogue of Somatic Mutations in Cancer) database [158]. Moreover, the ddPCR data highly correlate with the NGS results and meet the clinical need for a fast and cost-effective method for the detection of IDH mutations. 

In patients affected by AML, treated with conventional chemotherapy, it has been demonstrated that 2-HG levels progressively decrease concurrently with the reduction in tumor burden [159,160].

Among solid tumors, the first evidence of a favorable prognostic role of *IDH* mutations has been obtained in gliomas, where *IDH1* mutations are extremely frequent, accounting for more than 70% of previous WHO grade II and III astrocytomas and oligodendrogliomas, and in secondary glioblastomas [43]. 

In intrahepatic cholangiocarcinoma (iCCA), which is characterized by a high expression of 2-HG both in the tumor tissue and circulation, mostly related to the tumor burden, the prognostic data are controversial. Circulating 2-HG levels (with a threshold >170 ng/mL) could predict the presence of an IDHmut CCA with a sensitivity of 83% and a specificity of 90%, [161], but it is not predictive of clinical outcomes [162]. 

IDHmut low-grade or low-volume chondrosarcomas produced high levels of 2-HG when *IDH* mutations are present, but there is no correlation with the histopathological tumor grading. Moreover, IDHmut chondrosarcomas with higher intratumoral 2-HG levels at diagnosis, even in the absence of metastasis, have a worse OS, implying that intratumoral 2-HG may have a role as a prognostic biomarker [124]. Similarly, circulating 2-HG is elevated in IDHmut chondrosarcomas, but its role as a diagnostic biomarker is still debated because of its lower reliability when compared with the intratumoral value. This accuracy is reduced by the evidence that serum 2-HG levels seem to be higher in IDH wild-type chondrosarcoma than in other solid tumors and in healthy control patients, and there are currently no reports comparing peripheral blood 2-HG between the wild-type IDH chondrosarcoma and healthy populations [135].

Plasmatic IDH1 levels have also been evaluated as a potential biomarker of NSCLC; some studies have shown that plasmatic IDH1 levels are statistically significantly higher in NSCLC patients than in healthy controls [163]. Recently, the presence of the IDH2 protein in NSCLC patients’ serum has been demonstrated, observing that IDH2 protein levels were higher in patients compared to healthy controls. Moreover, the serum IDH2 protein levels decreased in patients with NSCLC at about one week after surgical removal of the tumor, suggesting a role as a diagnostic and prognostic biomarker; this could support the evaluation of the surgical outcome of patients with NSCLC [164].

Further studies are required to validate the utility of monitoring 2-HG serum levels in clinical practice as a surrogate biomarker in correlation with changes in tumor volume and the presence of metastatic disease but also as a potential pharmacodynamic marker of treatment response in IDHmut solid tumors.

### 3.2. IDH-Related Imaging, Spectroscopic, and Radiomics Biomarkers

Recent studies examined the possibility of associating the IDH phenotype in solid tumors, especially in gliomas, with specific radiological findings. The technological advancements in imaging, paired with high-performance computing and artificial intelligence, have revolutionized the role of imaging in early and noninvasive diagnosis, targeted treatment, and follow-up by providing access to vast amounts of data. For the initial diagnosis and staging of gliomas, the most relevant imaging modality is the traditional Magnetic Resonance Imaging (MRI). Moreover, the development of advanced MRI techniques such as diffusion, perfusion, and spectroscopy has led to the acquisition of microinvasive information that was previously undetectable by traditional MRI.

Recent studies focused on radiomics, which is defined as a high-throughput feature-extraction method able to unlock microscale quantitative data hidden within standard-of-care medical imaging. Another recent field of research is radiogenomics, which is defined as the linkage between imaging and genomic information. Multiple radiomics and radiogenomics studies performed on conventional and advanced neuro-oncology images show that they have the potential to differentiate pseudo-progression from true progression; classify tumor subgroups and grade; and predict recurrence, survival, and mutation status with high accuracy [165,166].

IDHmut glioma showed some distinctive features:Frontal lobe predominance with a low tendency to occupy high-risk brain regions such as the brainstem or diencephalon, which are typically related to IDH wild-type tumors, correlating with prognosis due to a higher chance of tumor resectability [167];Less contrast enhancement, suggesting a lower vascular permeability of the blood–brain barrier, usually disrupted by pathological tumor changes. However, enhanced regions in IDHmut gliomas are predictive of a worse outcome regarding PFS and OS, while IDH wild-type gliomas do not demonstrate a correlation between contrast enhancement (CE) and prognostic stratification [168];Well-defined borders, essential for radical tumor resection;“T2-FLAIR mismatch signs”, referring to regions on MRI presenting high signal intensity on a T2-weighted image but low intensity on Fluid-Attenuated Inversion Recovery (FLAIR) except for the hyperintense peripheral rim. However, interobserver variability is always an issue when applying qualitative image features, which radiomics strives to solve [169];High apparent diffusion coefficient (ADC) values, representing lower cellularity [170];Lower cerebral blood flow (CBV) values because IDHmut gliomas have low levels of HIF-1A via the 2-HG-mediated inhibition of Egg Laying Defective Nine protein (EGLN) and consequently show a decrease in proangiogenic signaling that is reflected as a lower CBV in perfusion-weighted MRI in comparison with the IDH wild-type [171].

Early imaging biomarkers such as FLAIR volume normalized relative to CBV (nrCBV), and ADC measurement can be usefully used for evaluating IDH inhibitor treatment response in human IDH1mut gliomas. Specifically, it was observed that the inhibition of IDH may increase vascularity as early as 3–6 weeks, and it leads to a transient increase in CBV that seems to stabilize after 2–4 months after the treatment. Moreover, PFS was strongly affected by changes in perfusion and ADC relative to baseline [172].

Additionally, the hallmark metabolic alterations of IDH-mutated gliomas can be analyzed by mass spectroscopy:Reduced lactate levels and near-normal intracellular pH in patients with IDHmut gliomas when comparing tumor voxels of patients with IDHmut with those of patients with IDH wild-type gliomas [173];Increased glutamate/glutamine before tumor shrinkage as potential translatable metabolic biomarkers of response to TMZ treatment in IDH1mut glioma [174];Overproduction of oncometabolite 2-HG, which plays a key role in malignant transformation; a decrease in 2-HG levels can be used to monitor a treatment’s early response in clinical trials of therapies targeting IDHmut [175].

Other studies focused on the determination of more complex parameters detected with Diffusion Tensor Imaging (DTI) and Diffusion Kurtosis Imaging (DKI), which are subsequential extensions of Diffusion Weighted Imaging (DWI) and show significant correlation with mutational status, high Ki-67 values, and a tendency towards a worse prognosis in glioma [176].

These findings suggest that the implementation of advanced MRI techniques in IDHmut gliomas may be performed as a noninvasive method to provide information about PFS and OS in this setting of patients. 

Recent studies have investigated the potential of O-(2-[18F]fluoroethyl)-L-tyrosine (FET) PET radiomics using textural features combined with static and dynamic parameters of FET uptake for the noninvasive prediction of IDH genotype. The presence of IDH mutations is associated with an increased expression of the amino acid transporter LAT1, which facilitates the accumulation of FET in tumor cells; thus, IDHmut gliomas may exhibit higher signal intensities on FET PET compared to IDH wild-type tumors. The highest diagnostic accuracy of 93% for a prediction of IDH genotype was achieved with the hybrid PET/MR scanner. An issue with this technique is that differences in the tumor volumes affected feature repeatability, significantly decreasing towards smaller VOIs (volumes of interest) [177].

Furthermore, patients with pseudo-progression showed a slightly lower and more homogenous FET uptake, whereas patients with early tumor progression showed a more heterogeneous FET uptake [178]. 

In conclusion, although they cannot replace histopathological characterization, prediction models based on radiomic features extracted from conventional MRI have shown promising results in identifying the characteristics of IDHmut tumors, particularly in gliomas [179].

## 4. Discussion

IDH1 and IDH2 are key metabolic enzymes that catalyze the conversion of isocitrate to α-ketoglutarate (α-KG). In recent years, some mutations in the *IDH* genes have been observed in several solid tumors such as glioma, cholangiocarcinoma, and chondrosarcoma. The inhibition of mutated IDH enzymes represents a promising treatment approach in solid tumors, with further development ongoing in current clinical trials [20,180,181,182].

The possibility of early diagnostics with noninvasive techniques (through advanced MRI or “liquid biopsy” with serum biomarkers such as 2-HG), which are currently applied in clinical practice only for gliomas [171,183], may become the standard approach for frequently IDH mutated tumors, with the use of standard tissue biopsy reserved to selected uncertain cases. IDH inhibitors may be an effective and manageable treatment for rare tumors with usually poor prognosis (e.g., SNUCs [141]) but also an alternative treatment in diseases where standard treatments may have a heavy burden of morbidities (such as radiotherapy-related cognitive impairment for gliomas).

Apart from solid malignancies which typically harbor IDH1/2 mutations as leading driver mutations (gliomas, iCCAs, chondrosarcoma, and SNUCs), many other solid tumors develop IDH mutations as a late event during disease progression [127,146,147,148,149,150]. In these cases, IDH inhibitors may be evaluated after the use of standard treatments. Considering that IDH inhibitors have been tested mostly in advanced or metastatic disease settings, it could be interesting to evaluate their efficacy in the context of early disease in IDH-mutated solid tumors. Moreover, IDH inhibitors have a clinical utility both as single agents and in combination with drugs that target different pathways (such as chemotherapy, immunotherapy, or other targeted therapies) [184]. The coexisting presence of epigenetic and metabolic alterations associated with IDH mutations has provided the rationale for testing drug combinations that target IDH enzymes and DNA repair or methylation/acetylation pathways. Although the resistance to anti-IDH1/2 drugs was observed in cholangiocarcinoma and could represent an obstacle to the long-term treatment of patients with IDH inhibitors, this problem may be overcome by the use of dual inhibitors (targeting both IDH1 and IDH2), which are currently under evaluation. 

## 5. Conclusions

IDH mutation acquisition is a relevant event in many tumor progressions, and targeting IDH mutations at an early stage of the disease can be crucial to halt disease progression and prevent the acquisition of additional genetic alterations that could reduce the effectiveness of IDH inhibitors. Great progress in the understanding of the role of IDH mutations has been observed in several cancers, and we expect an increase in the number of clinical trials aiming to identify the clinical benefit of IDH-targeted therapies and their combination strategies in IDHmut tumors.

## Figures and Tables

**Figure 1 cancers-16-02752-f001:**
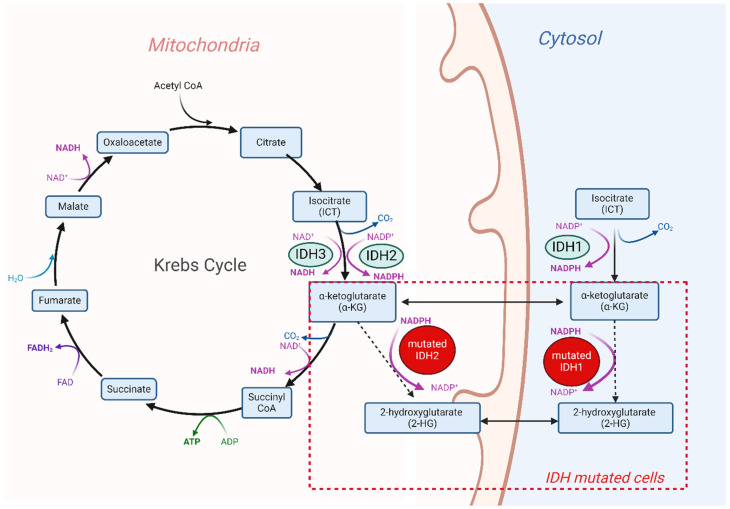
Physiological and pathological activity of the IDH enzymes in human cells. The IDH2 and IDH3 enzymes are located in mitochondria, and they are NADP+-dependent and NAD+-dependent enzymes, respectively. IDH1 is a NADP+-dependent enzyme and is distributed in the cytosol. Mutations in both the IDH1 and IDH2 enzyme genes are responsible for the conversion of α-ketoglutarate (α-KG) to 2-hydroxyglutarate (2-HG). Figure created with Biorender.com.

**Figure 2 cancers-16-02752-f002:**
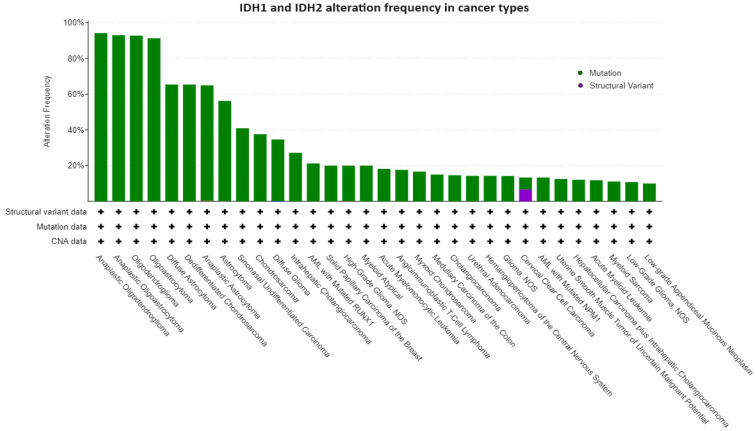
IDH1 and IDH2 alterations in cancer types. The figure shows the tumor types with higher alteration frequencies in the IDH1 and IDH2 genes. We filtered the tumor types with at least 10% of altered cases. The data and figures were imported from the GENIE Cohort v14.1 public dataset (all available samples with CNA and mutation on 12 December 2023, namely 147,940 samples/127,336 patients) available at https://genie.cbioportal.org/ (accessed on 12 December 2023).

**Figure 3 cancers-16-02752-f003:**
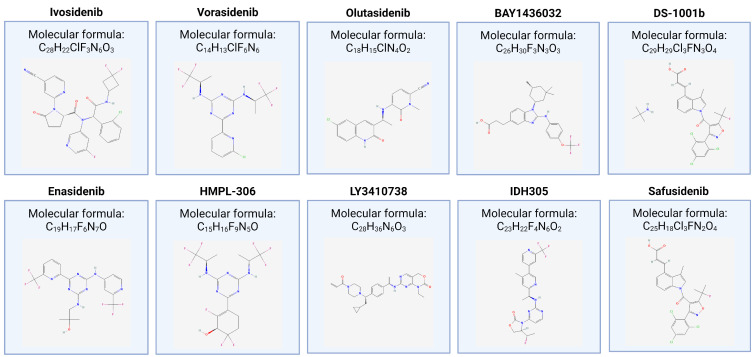
Chemical structures of the drug tested in completed and ongoing clinical trials that evaluate(d) IDH inhibitors for the treatment of glioma and cholangiocarcinoma. The 2D structure figures were obtained from https://pubchem.ncbi.nlm.nih.gov.

**Table 1 cancers-16-02752-t001:** IDH mutation prevalence and prognostic implication across various solid malignancies.

Disease	Gene	Mutation	Prevalence	Prognostic Implication
LGG/secondary GB	*IDH1*	IDH1 R132	>80%	Positive
iCCA	*IDH1*	IDH1 R132	20%	Unclear
Chondrosarcoma	*IDH1*, *IDH2*	IDH1 R132 IDH2 R172	50–60%	Positive (both)
Enchondroma	*IDH1*, *IDH2*	IDH1-R132C/H, IDH2-R172S	50%	Unclear
SNUCs	*IDH1*	IDH2-R172S/T	49%	Positive

LGG: low-grade glioma; GB: glioblastoma; iCCA: intrahepatic cholangiocarcinoma; SNUCs: sinonasal undifferentiated carcinomas.

**Table 3 cancers-16-02752-t003:** List of completed and ongoing clinical trials that evaluate(d) IDH inhibitors for the treatment of CCA. The IDH mutation types and their frequency are also reported.

Cholangiocarcinoma					
IDH Mutation Types				IDH1 R132X, IDH2 R172X.	
Frequency of IDH Mutations				40% (iCCA)	
Trial Name	Phase	Year	Drug Tested	Target Population	Outcome Measure
**Clinical Trials–Completed**					
NCT02989857 (ClarIDHy trial) [121]	III	2017–2021	Ivosidenib vs. placebo	Pre-treated advanced IDH1mut iCCA	mPFS 2.7 vs. 1.4 mo, mOS 10.8 vs. 9.7 mo
NCT03684811	Ib/II	2018–2022	Olutasidenib	Relapsed/refractory IDH1mut advanced solid tumors	ORR 12.5%
NCT04088188	I	2021–2023	Ivosidenib (combined with cisplatin/gemcitabine)	IDH1mut unresectable or metastatic CCA (Arm A)	mOS 22.9 mo, mPFS 15.4 mo
**Clinical Trials—Ongoing**					
NCT04521686 [122]	I	2020	LY3410738	IDH1mut or IDH2mut advanced solid tumors	
NCT04762602 [72]	I	2021	HMPL-306	IDHmut solid tumors	
NCT05814536	I	2023	Safusidenib	IDH1mut advanced CCA and other solid tumors	

iCCA: intrahepatic cholangiocarcinoma; mPFS: median progression-free survival; mo: months; mOS: median overall survival; ORR: objective response rate; CCA: cholangiocarcinoma.

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
