# Peer review of "Targeting Isocitrate Dehydrogenase (IDH) in Solid Tumors: Current Evidence and Future Perspectives"

_cancers, 2024, doi:10.3390/cancers16152752_

Round 1

Reviewer 1 Report

Comments and Suggestions for Authors

Dear authors,

Congratulation for your hard work that aims to shed a light in cancer metabolism via IDH, a potential therapeutic target.

Your work is comprehensive and well organized. Still, I would advise to revise the text because you have too many similarities with your cited references.

For example, all the paragraph between lines 477-482 is found, word by word in doi.org/10.1093/noajnl/vdac124 and two other sources. And it si not the only case. It would be a great idea to rephrase all the text taking into account your sources.

Author Response

Dear authors,
Congratulation for your hard work that aims to shed a light in cancer metabolism via IDH, a potential therapeutic target.Your work is comprehensive and well organized. Still, I would advise to revise the text because you have too many similarities with your cited references. For example, all the paragraph between lines 477-482 is found, word by word in doi.org/10.1093/noajnl/vdac124 and two other sources. And it is not the only case. It would be a great idea to rephrase all the text taking into account your sources.

Reply: We thank the reviewer for this observation. We checked the lines 477-482 and we agree that our paragraph had some similarities with the cited paper. We rephrased the indicated paragraph (now at lines 496-501) and other parts of the review as well (lines 23, 48-50, 55, 61-62, 72, 142-143, 210-211, 220-222, 459-460, 489).

Reviewer 2 Report

Comments and Suggestions for Authors

Carosi et al. present a review on targeting isocitrate dehydrogenase (IDH) in solid tumors. The paper provides an extensive overview of the current knowledge about IDH mutations in solid tumors. It highlights the significance of IDH1 and IDH2 mutations in various tumors, the promising approach of targeting these mutations with specific inhibitors, and the ongoing efforts to improve diagnostics and overcome treatment resistance through advanced strategies and combination therapies. Overall, this is an interesting and well-written article that is easy to read. The reviewer has only one suggestion: the authors may consider including the chemical structures of the drugs being evaluated. This addition would benefit readers interested in medicinal chemistry.

Author Response

Carosi et al. present a review on targeting isocitrate dehydrogenase (IDH) in solid tumors. The paper provides an extensive overview of the current knowledge about IDH mutations in solid tumors. It highlights the significance of IDH1 and IDH2 mutations in various tumors, the promising approach of targeting these mutations with specific inhibitors, and the ongoing efforts to improve diagnostics and overcome treatment resistance through advanced strategies and combination therapies. Overall, this is an interesting and well-written article that is easy to read. The reviewer has only one suggestion: the authors may consider including the chemical structures of the drugs being evaluated. This addition would benefit readers interested in medicinal chemistry.

Reply: We thank the reviewer for the comment. As suggested, we added in Figure 3 the chemical structures of the drugs listed in Table 2 and Table 3.

Reviewer 3 Report

Comments and Suggestions for Authors

The manuscript describes the current knowledge of IDHs as cancer targets and IDH inhibitors in the therapy of various cancers dependent on IDHs and its mutant forms. This work is informative, interesting and well-written. The length of the review is accurate. The provided figures and tables are correct and clear. However, I have some questions and suggestions the authors might care for in a revised version of the manuscript:

Several IDH inhibitors entered clinical trials for glioma therapy as mentioned in the manuscript. Can the IDH inhibitors replace other anti-glioma drugs such as azacytidine and temozolomide? Can synergy effects be expected in combination with these chemotherapeutics?

Because the combination with immune therapy seems to be very promising for cholangiocarcinoma therapy, would this combination also be promising for glioma therapy?

DS-1001b was mentioned with additional mechanisms of action. What are the advantages of other IDH inhibitors such as safusidenib and vorasidenib in comparison to ivosidenib?

Line 320: Please mention the names of the studied pan-HDAC inhibitors here in brackets.

Comments on the Quality of English Language

n.a.

Author Response

The manuscript describes the current knowledge of IDHs as cancer targets and IDH inhibitors in the therapy of various cancers dependent on IDHs and its mutant forms. This work is informative, interesting and well-written. The length of the review is accurate. The provided figures and tables are correct and clear. However, I have some questions and suggestions the authors might care for in a revised version of the manuscript:
Several IDH inhibitors entered clinical trials for glioma therapy as mentioned in the manuscript. Can the IDH inhibitors replace other anti-glioma drugs such as azacytidine and temozolomide? Can synergy effects be expected in combination with these chemotherapeutics?
Because the combination with immune therapy seems to be very promising for cholangiocarcinoma therapy, would this combination also be promising for glioma therapy?

DS-1001b was mentioned with additional mechanisms of action. What are the advantages of other IDH inhibitors such as safusidenib and vorasidenib in comparison to ivosidenib?
Line 320: Please mention the names of the studied pan-HDAC inhibitors here in brackets.

Reply: We thank the reviewer for this comment. We added a short paragraph about the use of combination therapy (chemotherapy and ivosidenib) in AML; there is currently less evidence about this strategy for IDHmut solid tumors. About immunotherapy, clinical trials are currently evaluating the potential role in gliomas; previous studies did not show a survival benefit. Ivosidenib and Safusidenib have a similar mechanism of action, but Ivosidenib has a broader spectrum of action. Indeed, while Safusidenib is more specific for IDH1 R132H and R132C, Vorasidenib is a dual IDH1/IDH2 inhibitor, with clinical activity both on IDH1 and IDH2 mutated gliomas. We added this information at lines 160-163, 171, 292-295. As suggested, we mentioned in brackets some of the 128 compounds among pan-HDAC inhibitors tested at line 320 (now line 335).

Reviewer 4 Report

Comments and Suggestions for Authors

Major Revisions

·         2-HG: there are D-2-HG and L-2-HG. Please clarify the stereoisomers and potentially differential effects of these two byproducts.

·         Role of 2-HG: It is recommended to add a description of the role of 2-HG in cancer development.

·         IDH1 vs. IDH2: their elevated expression in the TGCA database has differential effects on survival rate. Please add a description on the differences on mutations in IDH1 and IDHs.

Minor Revisions:

·         Table 1: please remove unnecessary space in each row.

·         Section 2.1: Please reduce the number of paragraphs in this subsection.

Author Response

Major Revisions • 2-HG: there are D-2-HG and L-2-HG. Please clarify the stereoisomers and potentially differential effects of these two byproducts. • Role of 2-HG: It is recommended to add a description of the role of 2-HG in cancer development. • IDH1 vs. IDH2: their elevated expression in the TGCA database has differential effects on survival rate. Please add a description on the differences on mutations in IDH1 and IDHs. Minor Revisions: • Table 1: please remove unnecessary space in each row. • Section 2.1: Please reduce the number of paragraphs in this subsection.

Reply: We thank the reviewer for this comment. We added a short explanation of the mechanisms of action of the two enantiomers of 2-HG and a more comprehensive description of the role of 2-HG in cancer development. We also added a paragraph about the observed difference in survival between IDH1 and IDH2 mutant gliomas. This information was added at lines 80-81, 90-91, 132-133. Finally, we corrected Table 1 and Section 2.1 according to the reviewer’s suggestion.

Reviewer 5 Report

Comments and Suggestions for Authors

The authors summarize the status of therapeutic development for IDH-mutated solid tumors by basic background and carcinoma. The review is well written and deserves to be published. However, the following points should be revised or added to improve the paper: 

1. L147-151. As the results of the Phase 1 trial of ivosidenib/vorasidenib for glioma, the reduction of tumor 1-HG concentration (92.6%) with vorasidenib was described, but those with the ivosidenib (91.1%) was not shown. The result for ivosidenib should also be included because this difference was the basis for vorasidenib to proceed to the Phase 3 study. 

2. L278-281. The standard care for cholangiocarcinoma is described, but the results of the main TOPAZ-1 trial were published in NEJM Evidence (DOI: 10.1056/EVIDoa2200015), which should be included as the reference. Moreover, pembrolizumab is also approved as an immune checkpoint inhibitor (Keynote-966, DOI: 10.1016/S0140-6736(23)00727-4) and should be listed as a standard therapy.

Author Response

The authors summarize the status of therapeutic development for IDH-mutated solid tumors by basic background and carcinoma. The review is well written and deserves to be published. However, the following points should be revised or added to improve the paper:

  1. L147-151. As the results of the Phase 1 trial of ivosidenib/vorasidenib for glioma, the reduction of tumor 1-HG concentration (92.6%) with vorasidenib was described, but those with the ivosidenib (91.1%) was not shown. The result for ivosidenib should also be included because this difference was the basis for vorasidenib to proceed to the Phase 3 study.
  2. L278-281. The standard care for cholangiocarcinoma is described, but the results of the main TOPAZ-1 trial were published in NEJM Evidence (DOI: 10.1056/EVIDoa2200015), which should be included as the reference. Moreover, pembrolizumab is also approved as an immune checkpoint inhibitor (Keynote-966, DOI: 10.1016/S0140-6736(23)00727-4) and should be listed as a standard therapy.

Reply: We thank the reviewer for the comment. We added the required information (lines 157 and 374) and the two references (117 and 135).